# Designing Reinforcement Learning Algorithms for Digital Interventions: Pre-Implementation Guidelines

**Anna L. Trella** [1,*], **Kelly W. Zhang** [1], **Inbal Nahum-Shani** [2], **Vivek Shetty** [3], **Finale Doshi-Velez** [1] **and Susan A. Murphy** [1]

1   School of Engineering and Applied Sciences, Harvard University, Cambridge, MA 02420, USA; kellywzhang@seas.harvard.edu (K.W.Z.); finale@seas.harvard.edu (F.D.-V.); samurphy@fas.harvard.edu (S.A.M.)
2   Institute for Social Research, University of Michigan, Ann Arbor, MI 48109, USA; inbal@umich.edu
3   Schools of Dentistry & Engineering, University of California, Los Angeles, CA 90095, USA; vshetty@ucla.edu
*   Correspondence: annatrella@g.harvard.edu

**Abstract:** Online reinforcement learning (RL) algorithms are increasingly used to personalize digital interventions in the fields of mobile health and online education. Common challenges in designing and testing an RL algorithm in these settings include ensuring the RL algorithm can learn and run stably under real-time constraints, and accounting for the complexity of the environment, e.g., a lack of accurate mechanistic models for the user dynamics. To guide how one can tackle these challenges, we extend the PCS (predictability, computability, stability) framework, a data science framework that incorporates best practices from machine learning and statistics in supervised learning to the design of RL algorithms for the digital interventions setting. Furthermore, we provide guidelines on how to design simulation environments, a crucial tool for evaluating RL candidate algorithms using the PCS framework. We show how we used the PCS framework to design an RL algorithm for Oralytics, a mobile health study aiming to improve users' tooth-brushing behaviors through the personalized delivery of intervention messages. Oralytics will go into the field in late 2022.

**Keywords:** reinforcement learning (RL); online learning; mobile health; algorithm design; algorithm evaluation

## 1. Introduction

There is growing interest in using online reinforcement learning (RL) to optimize the delivery of messages or other forms of prompts in digital interventions. In mobile health, RL algorithms have been used to increase the effectiveness of the content and timing of intervention messages designed to promote physical activity [1,2] and to manage weight loss [3]. In other areas, including the social sciences and education, RL algorithms are used to provide pretrial nudges to encourage court hearing attendance [4], to personalize math explanations [5], and to deliver quiz questions during lecture videos [6]. Unlike games and work in some areas of robotics, digital intervention studies can be extremely costly to run. Furthermore, when the study is a preregistered clinical trial, once initiated, the trial protocol (including any online algorithms) cannot be altered without jeopardizing trial validity. Thus, design decisions are a "one-way door" [7]; once we commit to a set of design decisions, they are irreversible for the duration of the trial. To prevent poor design decisions that could be detrimental to the effectiveness and the validity of study results, RL algorithms must undergo a thorough design and testing process before deployment.

The development of an RL algorithm for digital interventions requires a multitude of design decisions. These decisions include how best to accommodate the lack of mechanistic models for dynamic human responses to digital interventions and how to ensure the robustness of the algorithm to potentially nonstationary/non-Markovian outcome distributions. Furthermore, one must ensure not only that the RL algorithm learns and quickly

optimizes interventions but also that the algorithm runs stably and autonomously online within constrained amounts of time. One must also ensure that the RL algorithm can obtain data in a timely manner. Time and budgetary considerations may restrict the complexity of the RL algorithm that can be implemented. Furthermore, it is important to ensure the data collected by the RL algorithm can be used to inform future studies and address scientific, causal inference questions. Addressing these challenges in a reproducible, replicable manner is critical if RL algorithms are to play a role in optimizing digital interventions. Therefore, we need a framework for making design decisions for RL algorithms intended to optimize digital interventions.

The primary contributions of this work are twofold:

1. **Framework for Guiding Design Decisions in RL Algorithms for Digital Interventions:** *We provide a framework for evaluating the design of an online RL algorithm to increase confidence that the RL algorithm will improve the digital intervention's effectiveness in real-life implementation and maintain the intervention's reproducibility and replicability.* Specifically, we extend the PCS (predictability, computability, stability) data science framework of Yu [8] to address specific challenges in the development and evaluation of online RL algorithms for personalizing digital interventions.

2. **Case Study:** This case study concerns the development of an RL algorithm for Oralytics, a mobile health intervention study designed to encourage oral self-care behaviors. The study is planned to go into the field in late 2022. *This case study provides a concrete example of implementing the PCS framework to inform the design of an online RL algorithm.*

## 2. Review of Online Reinforcement Learning Algorithms

Reinforcement learning (RL) [9,10] is the area of machine learning that is concerned with learning how to best make a sequence of decisions. In digital intervention settings, the sequence of decisions concerns which treatment (e.g., motivational messages, reminders, types of feedback, etc.) to provide given the user's current state and history. Definitions of decision times, state, action, reward, and update times are provided below. The decision times and actions for the RL algorithm for Oralytics are provided in Figure 1.

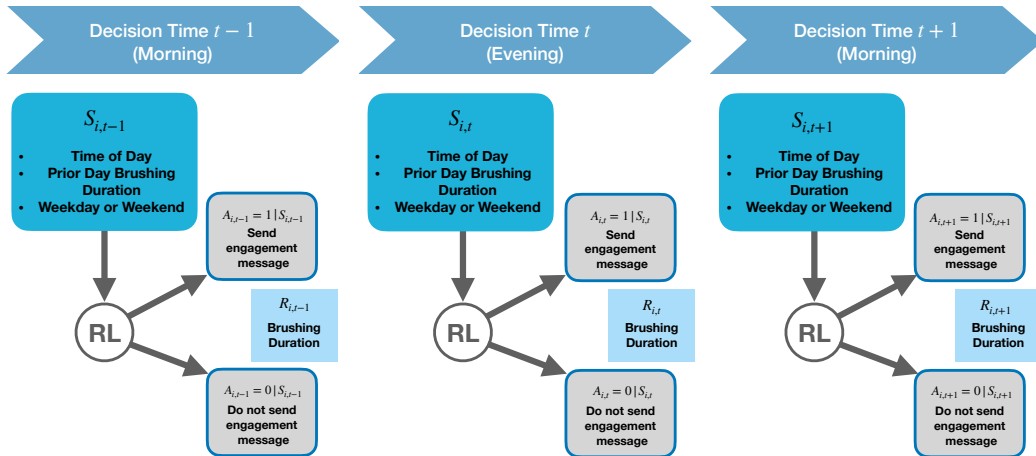

**Figure 1.** Decision times and actions for the RL algorithm in the Oralytics study. There will be two decision times per day (one in the morning and one in the evening); in total, each user will have 140 decision times, which we index by $t$. At a given decision time $t$, the RL algorithm receives state information for each user $S_{i,t}$ (this will include information on whether it is a morning or evening decision time and whether it is a weekday or weekend, as well as information on the user's previous day brushing). Given the state, the RL algorithm decides whether or not to send the user an engagement message (selects action $A_{i,t} \in \{0, 1\}$). After an action is taken, the RL algorithm receives reward $R_{i,t}$, which is the user's subsequent brushing duration in seconds during the brushing window following the decision time. See Section 5 for a discussion on the challenges of designing an RL algorithm for Oralytics.

### 2.1. Decision Times

These are the times, indexed by $t$, at which the RL algorithm may deliver a treatment (via a smart device such as a desktop computer, smartwatch, smartphone, smart speaker, wearable, etc.). The cadence of the decision times (minute level, hourly, daily, etc.) depends on the type of digital intervention. For example, in Oralytics, we have two decision times per day, namely, one hour prior to the set morning and evening brushing windows specified by the user.

### 2.2. State

$S_{i,t} \in \mathbb{R}^d$ represents the $i$th user's state at decision time $t$. $d$ is the number of features describing the user's state (e.g., current location, recent adherence to medication, current social setting, recent engagement with the intervention application, etc.). See Section 5.2 for the state definition for Oralytics.

### 2.3. Action

$A_{i,t} \in \mathcal{A}$ represents the decision made by the RL algorithm for the $i$th user at decision time $t$. Treatment actions in digital interventions frequently include the action of not delivering any treatment at time $t$. For Oralytics, the action space is $\mathcal{A} := \{0,1\}$, where $A_{i,t} = 1$ represents sending the user an engagement message and $A_{i,t} = 0$ represents not sending an engagement message. See Section 5.1 for descriptions of the types of messages that can be sent in Oralytics.

### 2.4. Reward

$R_{i,t} \in \mathbb{R}$ is the reward for the $i$th user at decision time $t$ observed after taking action $A_{i,t}$. The definition of the reward depends on the type of digital intervention. Examples include successfully completing a math problem, taking a medication, and increasing physical activity. In Oralytics, the reward is the subsequent brushing duration; see Section 5.2 for further discussion of the reward in Oralytics.

### 2.5. Online RL Algorithms

Generally, online RL algorithms are composed of two parts: (a) fitting a model of the user and (b) an action selection strategy. The simplest type of user model is a model for the *reward function*, $\mathbb{E}[R_{i,t}|S_{i,t}, A_{i,t}]$. In more general cases, a model for the sum of future rewards, conditional on the current state, $S_{i,t}$, and action, $A_{i,t}$, is also learned. The action selection strategy of the RL algorithm uses the user's current state $S_{i,t}$, along with the learned user model and outputs the treatment action $A_{i,t}$ at each decision time $t$.

### 2.6. Update Times

These are the times at which the RL algorithm is updated. Updating typically includes updating a model of the user (e.g., a model of the user's reward function). The RL algorithm updates using user $i$'s current history of past states, actions, and rewards up to time $t$, denoted by $H_{i,t-1} = \{S_{i,s}, A_{i,s}, R_{i,s}\}_{s=1}^{t-1}$. If the algorithm pools data across users, then the history of other users in the study, $H_{j,t-1}$ for $i \neq j$, is used to update the model for user $i$. These updates can occur after each decision time or at longer time scales. For example, in [1], the decision times are 5 times per day, but the update times are only nightly. For Oralytics, the update cadence is once a week.

An online RL algorithm should quickly learn which action to deliver in which states for each user. One of the most widely used and simplest RL algorithms is a contextual bandit algorithm [11–13]. As data accrues, a contextual bandit algorithm incrementally learns which action will lead to maximal reward in each state. The online algorithm sequentially updates its estimate of the reward function (the mean of the reward conditional on state and action) and selects actions. The performance of the algorithm is often measured by the sum of rewards—the faster the algorithm learns, the greater the sum.

## 3. PCS Framework for Designing RL Algorithms for Digital Intervention Development

The PCS Framework [8] incorporates best practices from machine learning and applied statistics to provide a set of guidelines for assessing the quality of a prediction algorithm when used to address problems in real life. The goal is to enhance the scientific community's confidence in the prediction algorithm's performance in terms of predictability of results, computability in the implementation of the algorithm, and stability in the performance results of the learning algorithm across perturbations. PCS has been adopted and extended to other domains; see Section 4.3 for further discussion.

As in the prediction setting, there are a variety of design decisions one needs to make before deploying an RL algorithm, e.g., choosing the model class to use to approximate the reward function. While many of the original PCS principles can be used in the development and evaluation of RL algorithms, RL algorithm development also introduces new challenges for the PCS framework, particularly in the online digital intervention setting. First, the main task is not prediction, but rather in RL, the main goal is to select intervention actions so that average rewards across time are maximized for each user. We call this the goal of personalization [14]. We generalize the PCS framework to include an evaluation of the ability of an online RL algorithm to personalize. Second, in digital intervention settings, it is important to evaluate the ability of the online RL algorithm to maximize rewards under real-world constraints. For example, there are often time constraints on computations, budgetary constraints on software engineering development, and constraints on the algorithm in terms of obtaining data in real time. Furthermore, the algorithm must run stably online without constant human monitoring and adjustment. The current PCS framework does not provide evaluation tools that deal with the above needs. We extend the PCS framework to the context of designing and evaluating online RL algorithms. Our extended framework focuses on providing confidence that the online RL algorithm will lead to greater effectiveness under real-world constraints and with stability.

### 3.1. Personalization (P)

The original PCS Framework uses predictability (P) to ensure that models used in data science have good predictive accuracy on both seen and unseen data. Predictive accuracy is a simple, commonly used metric for evaluating such models, but in some cases, multiple evaluation metrics or domain-specific metrics are more appropriate. In our setting, the main task is personalization. Namely, the online RL algorithm should learn to select actions to maximize *each* user's average rewards. Instead of a predictive accuracy metric, we want a metric to validate the extent of personalization. For example, when choosing a metric to evaluate RL algorithms for multiple users, one may be interested not just in the average over the users' sums of rewards but in other metrics that capture the variation in the sum of rewards across users. Let $N$ be the total number of users with $T$ total decision times. We suggest the following metrics:

- **Average of Users' Average (Across Time) Rewards:** This metric is the average of all $N$ users' rewards averaged across all $T$ decision times, defined as $\frac{1}{N} \sum_{i=1}^{N} \left( \frac{1}{T} \sum_{t=1}^{T} R_{i,t} \right)$. The metric serves as a global measure of the RL algorithm's performance.
- **The 25th Percentile of Users' Average (Across Time) Rewards:** To compute this metric, first compute the average reward across time for each user, $\frac{1}{T} \sum_{t=1}^{T} R_{i,t}$ for each $i = 1, 2, \ldots, N$; this metric is the lower 25th percentile of these average rewards across the $N$ users. The metric shows how well an RL algorithm performs for the worst-off users, namely users in the lower quartile of average rewards across time.
- **Average Reward For Multiple Time Points:** This metric is the average users' rewards across time for multiple time points $t_0 = 1, 2, ..., T$, defined as $\frac{1}{N} \sum_{i=1}^{N} \left( \frac{1}{t_0} \sum_{t=1}^{t_0} R_{i,t} \right)$ for each $t_0$. These metrics can be used to assess the speed at which the RL algorithm learns across weeks in the trial.

*3.2. Computability (C)*

Computability has to do with the efficiency and scalability of algorithms, decisions, and processes. While the original PCS framework focused on the computability of training and evaluating models, we also consider computability to include the ability to implement the algorithm within the constraints of the study. In the online RL setting, computability encompasses all issues related to ensuring that the RL algorithm can select actions and update in a timely manner while running online. The performance of the online RL algorithm must be evaluated under the constraints of the study; key RL algorithm design constraints that could arise include:

- **Timely Access to Reward and State Information:** The investigators may have an ideal definition of the reward or state features for the algorithm; however, due to delays in communication between sensors, the digital application, and the cloud storage, the investigators' first choice may not be reliably available. Since RL algorithms for digital interventions must make decisions online, the development team must choose state features that will be reliably available to the algorithm at each decision time. Additionally, the team must also choose rewards that are reliably available to the algorithm at update times.
- **Engineering Budget:** One should consider the engineering budget, supporting software needed, and time available to deliver a production-ready algorithm. If there are significant constraints, a simpler algorithm may be preferred over a sophisticated one because it is easier to implement, test, and set up monitoring systems for.
- **Off-Policy Evaluation and Causal Inference Considerations:** The investigative team often not only cares about the RL algorithm's ability to learn but also about being able to use data collected by the RL algorithm to answer scientific questions after the study is over. These scientific questions can include topics such as off-policy evaluation [15,16] and causal inference [17,18]. Thus, the algorithm may be constrained to select actions probabilistically with probabilities that are bounded away from zero and one. This enhances the ability of investigators to use the resulting data to address scientific questions with sufficient power [19].

*3.3. Stability (S)*

Stability concerns how an RL algorithm's results change with minor perturbations and the documentation and reproducibility of results. In online RL, stability plays two roles. First, the RL algorithm must run stably and automatically without the need for constant human monitoring and adjustment. This is particularly critical as users abandon digital interventions that have inconsistent functionality (unstable RL algorithm) [20,21]. Second, the RL algorithm should perform well across a variety of potential real-world environments. A critical tool in assessing stability to perturbations of the environment is the use of simulation test beds. Test beds include a variety of plausible environmental variants, each of which encodes different concerns of the investigative team. The following are challenging attributes of probable environments in digital intervention problems that one could design test beds for:

- **User Heterogeneity:** There is likely some amount of user heterogeneity in response to actions, even when users are in the same context. User heterogeneity can be partially due to unobserved user traits (e.g., factors that are stable or change slowly over time, like family composition or personality type). The amount of between-user heterogeneity impacts whether an RL algorithm that pools data (partially or using clusters) across users to select actions will lead to improved rewards.
- **Non-Stationarity:** Unobserved factors common to all users such as societal changes (e.g., a new wave of the pandemic), and time-varying unobserved treatment burden (e.g., a user's response to a digital intervention may depend on how many days the user has experienced the intervention) may make the distribution of the reward appear to vary with time, i.e., non-stationary.

- **High-Noise Environments:** Digital interventions typically deliver treatments to users in highly noisy environments. This is in part because digital interventions deliver treatments to users in daily life, where many unobserved factors (e.g., social context, mood, or stress) can affect a user's responsiveness to an intervention. If unobserved, these factors produce noise. Moreover, the effect of digital prompts on a near-term reward tends to be small due to the nature of the intervention. Therefore, it is important to evaluate the algorithm's ability to personalize even in highly noisy, low signal-to-noise ratio environments.

*3.4. Simulation Environments for PCS Evaluation*

To utilize the PCS framework, we advocate for using a simulation environment for designing and evaluating RL algorithms. We aim to compare RL algorithm candidates under real-world constraints (computability). Thus, we build multiple variants of the simulation environment, each reflecting plausible user dynamics (i.e., state transitions and reward distributions) (stability). We then simulate digital intervention studies for each simulation environment variant and RL algorithm candidate pairing. Finally, we use multiple metrics to evaluate the performance of the RL algorithm candidates (personalization).

In the case of digital interventions, there is often no mechanistic model or physical process for user behavioral dynamics, which makes it difficult to accurately model transitions (e.g., modeling a user's future level of physical activity as a function of their past physical activity, location, local weather). Note that the goal of developing the simulators is not to conduct model-based RL [22]. Rather, here, the simulators represent a variety of plausible environments to facilitate the evaluation of the performance of potential RL algorithms in terms of personalization, computability, and stability across these environments. Existing data and domain expertise is most naturally used to construct the simulation environments. However, as is the case for Oralytics, the previously collected data may be scarce, i.e., we have few data points per user. Moreover, the data may only be partially informative, e.g., the data was collected under only a subset of the actions. Next, we provide guidelines for how to build an environment simulator in such challenging settings.

Base Environment Simulator: To have the best chance possible of accurately evaluating how well different RL candidates will perform, we recommend first building a base environment simulator that mimics the existing data to the greatest extent possible. This involves carefully choosing a set of time-varying features and reward-generating model class that will be expressive enough to model the true reward distribution well. To check how well the simulated data generated by the model of the environment mimics the observed data, we recommend a variety of ways to compare distributions. This includes visual comparisons such as plotting histograms; comparing measures of the real data such as mean reward, between-user variance, and within-user variance to the same measures of the simulated data; and measuring how well the base model captured the variance in the data. Examples of these checks done for Oralytics are in Appendix A.4.

Variant Environment Simulators: We recommend considering many variants or perturbed simulation environments to evaluate the stability of RL algorithms across multiple plausible environments. These variants can be used to address the concerns of the investigative team. For example, if the base simulator generates stationary rewards and the investigative team is concerned that the real reward distribution may not be stationary, a variant could incorporate nonstationarity into the environment dynamics.

If the previously collected data does not include particular actions, as was the case for Oralytics, we recommend consulting domain experts for a range of potential realistic effect sizes (differences in mean reward under the new action versus a baseline action). For example, in Oralytics, we only have data under no intervention and do not have data on rewards under the intervention. Thus, using the input of the domain experts on the team, we imputed several plausible treatment effects (varying by certain state features and the amount of heterogeneity in treatment effects across users).

## 4. Related Works

### 4.1. Digital Intervention Case Studies

Liao et al. [1] describes the development of an online RL algorithm for HeartSteps V2, a physical activity mobile health application. The authors highlight how their design decisions address specific challenges in designing RL algorithms by, for example, adjusting for longer-term effects of the current action and accommodating noisy data. However, they do not provide general guidelines for making design decisions for RL algorithms in digital intervention development.

Another related work is that of Figueroa et al. [23], which provides an in-depth case study of the design decisions, and the associated challenges and considerations, for an RL algorithm for text messaging in a physical activity mobile application serving patients with diabetes and depression. This case study provides guidelines to others developing RL algorithms for mobile health applications. Specifically, the authors first categorize the challenges they faced into 3 major themes: (1) choosing a model for decision making, (2) data handling, and (3) weighing algorithm performance versus effectiveness in the real world. They describe how they dealt with each challenge in the design process of their RL algorithm. In contrast, by expanding the PCS framework, this work introduces general guidelines for comprehensively evaluating RL algorithms. Moreover, we make recommendations for how to design a variety of simulation test beds even using only sparse and partially informative existing data, in service of PCS. The generality of the PCS framework makes it more widely applicable. For example, Figueroa et al. [23] has an existing dataset for all actions, which makes its recommendations less applicable to those designing algorithms with existing data only under a subset of actions. The PCS framework allows us to move beyond suggesting solutions to a specific set of challenges for a particular study by offering holistic guidelines for addressing challenges in developing simulation environments and evaluating algorithms.

### 4.2. Simulation Environments in Reinforcement Learning

In RL, simulators (generative models) may be used to derive a policy from the generative model underlying the simulator (model-based learning). Agarwal et al. [22] uses simulation as an intermediate step to learn personalized policies in a data-sparse regime with heterogeneous users, where they only observe a single trajectory per user. Wei et al. [24] proposes a framework for simulating in a data-sparse setting by using imitation learning to better interpolate traffic trajectories in an autonomous driving setting. In contrast, in PCS, the simulator is used as a crucial tool for using the framework to design, compare, and evaluate RL algorithm candidates for use in a particular problem setting.

There exist many resources aiming to improve the design and evaluation of RL algorithms through simulation; however, in contrast to this work, they do not provide guidelines for designing plausible simulation environments using existing data. RecSim [25] gives a general framework but does not advise on the quality of the environment nor on how to make critical design decisions such as reward construction, defining the state space, simulating unobserved actions, etc. MARS-Gym [26] provides a full end-to-end pipeline process (data processing, model design, optimization, evaluation) and open-source code for a simulation environment for marketplace recommender systems. OpenAI Gym [27] is a collection of benchmark environments in classical control, games, and robotics where the public can run and compare the performance of RL algorithms.

There are also a handful of papers that build simulation environment test beds using real data. Wang et al. [28] evaluates their algorithm for promoting running activity with a simulation environment built using two datasets. Singh et al. [29] develops a simulation environment using movie recommendations to evaluate their safe RL approach. Korzepa et al. [30] uses a simulation environment to guide the design of personalized algorithms that optimize hearing aid settings. Hassouni et al. [31,32] fits a realistic simulation environment using the U.S. timekeeping research project data. Their simulation environment creates daily schedules of activities for each user (i.e., sleep, work, workout, etc.)

where each user is one of many different user profiles (i.e., workaholic, athlete, retiree) for the task of improving physical activity.

*4.3. PCS Framework Extensions*

The PCS framework has been extended to other learning domains such as causal inference [33], network analysis [34], and algorithm interpretability [35]. Despite the variety of these tasks, they can all be framed as supervised learning problems in batch data settings that can be evaluated in terms of prediction accuracy on a hold-out dataset. PCS has not been extended to provide guidelines for developing an online decision-making algorithm. This extension is needed because of the additional considerations, discussed above, present in a real-world RL setting. Additionally, while these papers focus on evaluating how well a model accurately predicts the outcome on training and hold-out datasets, we extend the framework to evaluate how well an algorithm personalizes to each user. Dwivedi et al. and Ward et al. [33,34] implement the original computability principle by considering algorithm and process efficiency and scalability. Margot et al. [35] provides a new principle, simplicity, which is based on the sum of the lengths of generated rules. We extend computability to include the constraints of the study. Finally, these papers consider the stability of results across different changes to the data (e.g., bootstrapping or cross-validation) or design decisions (e.g., choice of representation space or the embedding dimension). Our framework focuses on how stable an algorithm is in plausible real-world environments that may be complex (e.g., due to user heterogeneity, nonstationary, high noise).

**5. Case Study: Oral Health**

In this case study, we demonstrate the use of PCS principles in designing an RL algorithm for Oralytics. Two main challenges are (1) we do not have timely access to many features and the reward is relatively noisy and (2) we have sparse, partially informative data to inform the construction of our simulation environment test bed. In addition, there are several study constraints.

1. Once the study is initiated, the trial protocol and algorithm cannot be altered without jeopardizing trial validity.
2. We are using an online algorithm, so we may not have timely access to certain desirable state features or rewards.
3. We have a limited engineering budget.
4. We must answer post-study scientific questions that require causal inference or off-policy evaluation.

We highlight how we handle these challenges by using the PCS framework, despite being in a highly constrained setting. The case study is organized as follows. In Section 5.1, we give background context and motivation for the Oralytics study. In Section 5.2, we explain the Oralytics sequential decision-making problem. In Section 5.3, we describe our process for designing RL algorithm candidates that can stably learn despite having a severely constrained features space and noisy rewards. Finally, in Section 5.4, we describe how we designed the simulation environment variants to evaluate the RL algorithm candidates; throughout, we offer recommendations for designing realistic environment variants and for constructing such environments using data for only a subset of actions.

*5.1. Oralytics*

Oralytics is a digital intervention for improving oral health. Each user is provided a commercially available electric toothbrush with integrated sensors and Bluetooth connectivity as well as the Oralytics mobile application for their smartphone. There are two decision times per day (prior to the user's morning and evening brushing windows) when a message may or may not be delivered to the user via their smartphone. The types of messages focus on winning a gift for oneself, winning a gift for one's favorite charity, feedback on prior brushing, and educational information. Once a message is delivered to the user, the app records it so that a user is highly unlikely to receive the same message

twice. Oralytics will be implemented with approximately 70 users in a clinical trial where the participant duration is 10 weeks; this means each user has $T = 140$ decision times. The study duration is 2 years and the expected weekly incremental recruitment rate is around 4 users. The Oralytics mobile app will use an online RL algorithm to optimize message delivery (i.e., treatment actions) to maximize an oral health-related reward (see below). To inform the RL algorithm design, we have access to data from a prior oral health study, ROBAS 2 [36], and input from experts in oral and behavioral health. The ROBAS 2 study used earlier versions of both the electric toothbrush and the Oralytics application to track the brushing behaviors of 32 users over 28 days. Importantly, in ROBAS 2, no intervention messages were sent to the users.

*5.2. The Oralytics Sequential Decision-Making Problem*

We now discuss how we designed the state space and rewards for our RL problem in collaboration with domain experts and the software team while considering various constraints. These decisions must be communicated and agreed upon with the software development team because they build the systems that provide the RL algorithm with the necessary data at decision and update times and execute actions selected by the RL algorithm.

1. Choice of Decision Times: We chose the decision times to be prior to each user's specified morning and evening brushing windows, as the scientific team thought this would be the best time to influence users' brushing behavior.

2. Choice of Reward: The research team's first choice of reward was a measure of brushing quality derived from the toothbrush sensor data from each brushing episode. However, the brushing quality outcome is often not reliably obtainable because it requires (1) that the toothbrush dock be plugged in and (2) that the user be standing within a few feet of the toothbrush dock when brushing their teeth. Users may fail to meet these two requirements for a variety of reasons, e.g., the user brushes their teeth in a shared bathroom where they cannot conveniently leave the dock plugged in. Thus, we selected brushing duration in seconds as the reward (personalization) since 120 s is the dentist-recommended brushing duration and brushing duration is a necessary factor in calculating the brushing quality score. Additionally, brushing duration is expected to be reliably obtainable even when the user is far from the toothbrush dock when brushing (computability). Note that in Figure 2, a small number of user-brushing episodes have durations over the recommended 120 s. Hence, we truncate the brushing time to avoid optimizing for overbrushing. Let $D_{i,t}$ denote the user's brushing duration. The reward is defined as $R_{i,t} := \min(D_{i,t}, 180)$.

3. Choice of State Features At Decision Time: To provide the best personalization, an RL algorithm ideally has access to as many relevant state features as possible to inform a decision, e.g., recent brushing, location, user's schedule, etc. However, our choice of the state space is constrained by the need to get features reliably before decision and update times, as well as by our limited engineering budget. For example, we originally wanted a feature for the evening decision time to be the morning's brushing outcome; however, this feature may not be accessible in a timely manner. This is because in order for the algorithm to receive the morning brushing data, the Oralytics smartphone app requires the user to open the app and we do not expect most users to reliably open the app after every morning brush time before the evening brushing window. Further discussion of our choice of decision time state features can be found in Appendix B.1.

4. Choice of Algorithm Update Times: In our simulations, we update the algorithm weekly. In terms of speed of learning (at least in idealized settings), it is best to update the algorithm after each decision time. However, due to computability considerations, we chose a slower update cadence. Specifically, for the Oralytics app, the consideration was that we can only update the policy used to select actions when the user opens the app. If the user did not open the app for many days, we would be unable to update the app after each decision time. Users may well fail to open the app for a few days at a time, so we

chose weekly updates. In the future, we will explore other update cadences as well, e.g., once a day.

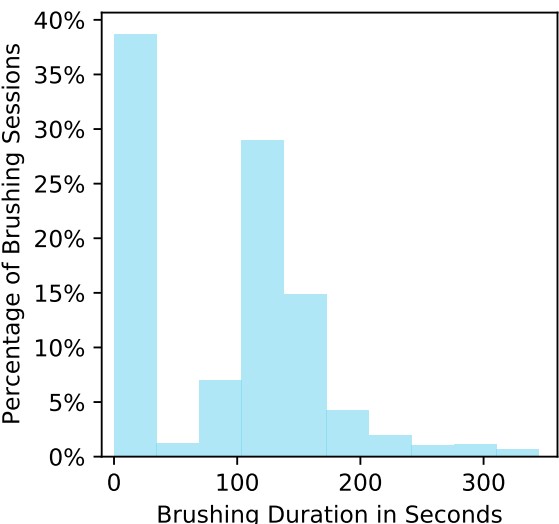

**Figure 2.** Histogram of brushing durations in seconds for all user brushing sessions in ROBAS 2. The ROBAS 2 study had 32 users total and each user had 56 brushing windows (2 brushing windows per day for 28 days). If a user did not brush during a brushing window, their brushing duration is recorded as zero seconds. Note in the figure above that across all users and brushing windows, about 40% of brushing sessions had no brushing, that is, a brushing duration of zero seconds. The ROBAS 2 brushing durations are highly zero-inflated.

### 5.3. Designing the RL Algorithm Candidates

Here, we discuss our use of the PCS framework to guide and evaluate the following design decisions for the RL algorithm candidates. There are some decisions that we have already made and other decisions that we encode as axes for our algorithm candidates to test in the simulation environment. See Appendix B for further details regarding the RL algorithm candidates.

1. Choice of using a Contextual Bandit Algorithm Framework: We understand that actions will likely affect a user's future states and rewards, e.g., sending an intervention message the previous day may affect how receptive a user is to an intervention message today. This suggests that an RL algorithm that models a full Markov decision process (MDP) may be more suitable than a contextual bandit algorithm. However, the highly noisy environment and the limited data to learn from (140 decision times per user total) make it difficult for the RL algorithm to accurately model state transitions. Due to errors in the state transition model, the estimates of the delayed effects of actions used in MDP-based RL algorithms can often be highly noisy or inaccurate. This issue is exacerbated by our severely constrained state space (i.e., we have few features and the features we get are relatively noisy). As a result, an RL algorithm that fits a full MDP model may not learn much during the study, which could compromise personalization and offer a poor user experience. To mitigate these issues, we use contextual bandit algorithms, which fit a simpler model of the environment. Using a lower discount factor (a form of regularization) has been shown to lead to learning a better policy than using the true discount factor, especially in data-scarce settings [37]. Thus, a contextual bandit algorithm can be interpreted as an extreme form of this regularization where the discount factor is zero. Finally, contextual bandits are the simplest algorithm for sequential decision making (computability) and have been used to personalize digital interventions in a variety of areas [1,2,5,23].

2. Choice of a Bayesian Framework: We consider contextual bandit algorithms that use a Bayesian framework, specifically posterior (Thompson) sampling algorithms [38]. Posterior sampling involves placing a prior on the parameters of the reward approximating

function and updating the posterior distribution of the reward function parameters at each algorithm update time. This allows us to incorporate prior data and domain expertise into the initialization of the algorithm parameters. In addition, Thompson sampling algorithms are stochastic (action selections are a not deterministic function of the data), which better facilitate causal inference analyses later on using the data collected in the study.

3. Choice of Constrained Action Selection Probabilities: We constrain the action selection probabilities to be bounded away from zero and one in order to facilitate off-policy and causal inference analyses once the study is over (computability). With help from the domain experts, we decided to constrain the action selection probabilities of the algorithm to be in the interval $[0.35, 0.75]$.

The following are decisions we will test using the simulation environment.

4. Choice of the Reward Approximating Function: An important decision in designing the contextual bandit algorithm is how to approximate the reward function. We consider two types of approximations, a Bayesian linear regression model (BLR) and a Bayesian zero-inflated Poisson regression model (ZIP), which are both relatively simple, well studied, and well understood. For BLR, we implement action centering in the linear model [1]. The linear model for the reward function is easily interpretable by domain experts and allows them to critique and inform the model. We consider the ZIP because of the zero-inflated nature of brushing durations in our existing dataset ROBAS 2; see Figure 2. We expect the ZIP to provide a better fit to the reward function by the contextual bandit and thus lead to increased average rewards. Formal specifications for BLR and ZIP as reward functions can be found in Appendix B.2.1 and Appendix B.2.2, respectively.

To perform posterior sampling, both the BLR and ZIP models are Bayesian with uninformative priors. From the perspective of computability and stability, the posterior for the BLR has a closed form, which makes it easier to write software that performs efficient and stable updates. In contrast, for the ZIP, the posterior distribution must be approximated and the approach used to approximate the posterior is another aspect of the algorithm design that the scientific team needs to consider. See Appendix C for further discussion on how to update the RL algorithm candidates.

5. Choice of Cluster Size: We consider clustering users with cluster sizes $K = 1$ (no pooling), $K = 4$ (partial pooling), and $K = N = 72$ (full pooling) to determine whether clustering in our setting will lead to higher sums of rewards (personalization). Note that 72 is the approximate expected sample size for the Oralytics study. Clustering-based algorithms pool data from multiple users to learn an algorithm per cluster (i.e., at update times, the algorithm uses $H_{i,t-1}$ for all users $i$ in the same cluster, and at decision times, the same algorithm is used to select actions for all users in the cluster). Clustering-based algorithms have been empirically shown to perform well when users within a cluster are similar [39,40]. In addition, we believe that clustering will facilitate learning within environments that have noisy within-user rewards [41,42]. There is a trade-off between no pooling and full pooling. No pooling may learn a policy more specific to the user later on in the study but may not learn as well earlier in the study when there is not a lot of data for that user. Full pooling may learn well earlier in the study because it can take advantage of all users' data but may not personalize as well as a no-pooling algorithm, especially if users are heterogeneous. We consider $K = 4$ for the balance partial pooling offers between the advantages and disadvantages between no pooling and full pooling. Moreover, four is the study's expected weekly recruitment rate and the update cadence is also weekly. We consider the two extremes and partial pooling as a way to explore this trade-off. A further discussion on choices of cluster size can be found in Appendix B.3.

### 5.4. Designing the Simulation Environment

We build a simulator that considers multiple variants for the environment, each encoding a concern by the research team. The simulator allows us to evaluate the stability of results for each RL algorithm across the environmental variants (stability).

Fitting Base Models: Recall that the ROBAS 2 study did not involve intervention messages. However, we can still use the ROBAS 2 dataset to fit the base model for the simulation environment, i.e., a model for the reward (brushing duration in seconds) under no action. Two main approaches for fitting zero-inflated data are the zero-inflated model and the hurdle model [43]. Both the zero-inflated model and the hurdle model have (i) a Bernoulli component and (ii) a nonzero component. The zero-inflated model's Bernoulli component is latent and represents the user's intention to brush, while the hurdle model's Bernoulli component is observed and represents whether the user brushed or not. Therefore, the zero-inflated model's nonzero component models the user's brushing duration when the user intends to brush, and the hurdle model's nonzero component models the user's brushing duration conditional on whether the user brushed or not. Throughout the model fitting process, we performed various checks on the quality of the model to determine whether the fitted model was sufficient (Appendix A.4). This included checking whether the percentage of zero brush times simulated by our model was comparable to that of the original ROBAS 2 dataset. Additionally, we checked whether the model accurately captured the mean and variance of the nonzero brushing durations across users.

The first approach we took was to choose one model class (zero-inflated Poisson) and fit a single population-level model for all users in the ROBAS 2 study. However, a single population-level model was insufficient for fitting all users due to the high level of user heterogeneity (i.e., the between-user and within-user variance of the simulated brushing durations from the fitted model was smaller than the between-user and within-user variance of brushing durations in the ROBAS 2 data). Thus, next, we decided to maintain one model class, but fit one model per user for all users. However, when we fit a zero-inflated Poisson to each user, we found that the model provided an adequate fit for some users but not for users who showed more variability in their brushing durations. The within-user variance simulated rewards from the model fit on those users was still lower than the within-user variance of the ROBAS 2 user data used to fit the model. Therefore, we considered a hurdle model [43] because it is more flexible than the zero-inflated Poisson. For Poisson distributions, the mean and variance are equal, whereas the hurdle model does not conflate the mean and variance.

Ultimately, for each user, we considered three model classes: (1) a zero-inflated Poisson, (2) a hurdle model with a square root transform, and (3) a hurdle model with a log transform, and chose one of these model classes for each user (Appendix A.2). Specifically, to select the model class for user $i$, we fit all three model classes using each user's data from ROBAS 2. Then, we chose the model class that had the lowest root mean squared error (RMSE) (Appendix A.3). Additionally, along with the base model that generates stationary rewards, we include an environmental variant with a nonstationary reward function; here, "day in study" is used as a feature in the environment's reward generating model (Appendix A.1).

Imputing Treatment Effect Sizes: To construct a model of rewards for when an intervention message is sent (a case for which we have no data), we impute plausible treatment effects with the interdisciplinary team and modify the fitted base model with these effects. Specifically, we impute treatment effects on the Bernoulli component and the nonzero component. We impute both types of treatment effects because the investigative team's intervention messages were developed to encourage users to brush more frequently and to brush for the recommended duration. Furthermore, because the research team believes that the users may respond differently to the engagement messages depending on the context and depending on the user, we included context-aware, population-level, and user-heterogeneous effects of the engagement messages as environmental variants (Appendix A.5).

We use the following guidelines to guide the design of the effect sizes:

1.    In general, for mobile health digital interventions, we expect the effect (magnitude of weight) of actions to be smaller than (or on the order of) the effect for baseline features, which include time of day and the user's previous day brushing duration (all features are specified in Appendix A.1).

2.  The variance in treatment effects (weights representing the effect of actions) across users should be on the order of the variance in the effect of features across users (i.e., variance in parameters of fitted user-specific models).

Following guideline 1 above, to set the population level effect size, we take the absolute value of the weights (excluding that for the intercept term) of the base models fitted for each ROBAS 2 user and the average across users and features (e.g., the average absolute value of weight for time of day and previous day brushing duration). For the heterogeneous (user-specific) effect sizes, for each user, we draw a value from a normal centered at the population effect sizes. Following guideline 2, the variance of the normal distributions is found by again taking the absolute value of the weights of the base models fitted for each user, averaging the weights across features, and taking the empirical variance across users. In total, there are eight environment variants, which are summarized in Table 1. See Appendix A for further details regarding the development of the simulation environments.

**Table 1.** Four Environment Variants. We consider two environment base models (stationary and nonstationary) and two effect sizes (population effect size, heterogeneous effect size).

| | |
|---|---|
| **S_Pop:** Stationary Base Model, Population Effect Size | **NS_Pop:** Nonstationary Base Model, Population Effect Sizes |
| **S_Het:** Stationary Base Model, Heterogeneous Effect Size | **NS_Het:** Nonstationary Base Model, Heterogeneous Effect Sizes |

## 6. Experiment and Results

We evaluate the RL algorithm candidates in each of the environment variants (stability). Specifically, the RL algorithm candidates will be comprised of a posterior sampling algorithm with two different reward models: (i) a Bayesian linear regression model (BLR) and (ii) a zero-inflated Poisson regression model (ZIP); see Appendices B and C for more discussion of these algorithms. Additionally, for each of these two reward approximating functions, we will consider different cluster sizes (with $k = 1, 4, N$ users). Each cluster will have one RL algorithm instantiation per cluster (no data shared across clusters). We cluster users by their entry date into the study (e.g., the first $k$ users are in the first cluster, the next $k$ users are in the second cluster, and so on). Since for the real Oralytics study we will incrementally recruit users into the study at a rate of about four users per week, for our experiments, we also simulate four users entering the study every week. To simulate a study, we draw $N = 72$ users (approximately the expected sample size for the Oralytics study) with replacement and cluster them by their entry date into the simulated study. The algorithms for each cluster are updated weekly with the first update taking place after one week (at decision time $t = 14$ for each cluster).

To evaluate personalization, we use the following metrics to compare algorithms: average rewards (average across users and time) and the 25th percentile of average rewards (averaged over time) across users. The purpose of looking at the 25th percentile of average rewards across users is to evaluate how well the algorithms perform on the worst off users (i.e., users who have a lower average reward than the average user). We ran 100 Monte Carlo trials for each environmental variant and algorithm candidate pairing. Table 2 shows the average and the 25th percentile of users' average (across time) rewards. Figure 3 shows the average reward over time.

We highlight the following takeaways from our experiments:

1.  **BLR vs. ZIP:** We prefer BLR to ZIP. BLR with cluster size $k = N$ results in higher user rewards than all other RL algorithm candidates in all environments in terms of average reward and 25th percentile reward (Table 2) and for average reward across all user decision times (Figure 3). It is interesting to note that BLR with cluster size $k = 4$ performs comparably to ZIP for all cluster sizes $k$ (Table 2, Figure 3). Originally, we hypothesized that ZIP would perform better than BLR because the ZIP-based algorithms can better model the zero-inflated nature of the rewards. We believe that

the ZIP-based algorithms suffered in performance because they require fitting more parameters and thus require more data to learn effectively. On the other hand, the BLR model trades off bias and variance more effectively in our data-sparse settings. Beyond considerations of their ability to personalize, we also prefer the BLR-based RL algorithms because they have an easy-to-compute closed-form posterior update (computability and stability). The ZIP-based algorithms involve using approximate posterior sampling, which is more computationally intensive and numerically unstable. In addition, BLR with action centering is robust, namely, it is guaranteed to be unbiased even when the baseline reward model is incorrect [1]. BLR with action centering specifically does not require the knowledge of the baseline features at decision time (See Appendix C.1.1). This means that baseline features only need to be available at update time and we can incorporate more features that were not available in real time at the decision time.

**Table 2.** Average and 25th Percentile Rewards. Average and 25th percentile rewards are defined in Section 3.1. The naming convention for environment variants is found in Table 1. "k" refers to the cluster size. Average rewards are averaged across time, users, and 100 trials. For the 25th percentile rewards, we average rewards for each user across time, find the lower 25th percentile across $N = 72$ users, and then average that across 100 trials. The value in the parenthesis is the standard error of the mean. The best performing algorithm candidate in each environment variant is bolded. BLR ($k = N$) performs better than other algorithm candidates across all simulated environments. Notice that the average rewards are lower than the 120-s dentist-recommended brushing duration. This is because of the zero-inflated nature of our setting (i.e., the user does not brush).

| RL Algorithm Candidates | | | | |
|---|---|---|---|---|
| Average Rewards | | | | |
| RL Algorithm | S_Het | NS_Het | S_Pop | NS_Pop |
| ZIP $k = 1$ | 100.038 (0.597) | 102.566 (0.526) | 107.184 (0.626) | 109.379 (0.552) |
| ZIP $k = 4$ | 100.463 (0.586) | 103.035 (0.539) | 108.217 (0.609) | 110.242 (0.562) |
| ZIP $k = N$ | 100.791 (0.596) | 103.391 (0.546) | 108.410 (0.617) | 110.542 (0.554) |
| BLR $k = 1$ | 97.196 (0.585) | 99.691 (0.527) | 103.692 (0.615) | 105.590 (0.546) |
| BLR $k = 4$ | 99.772 (0.590) | 102.310 (0.547) | 107.568 (0.619) | 109.454 (0.547) |
| BLR $k = N$ | **101.267 (0.590)** | **104.024 (0.542)** | **108.974 (0.610)** | **111.201 (0.546)** |
| 25th Percentile Rewards | | | | |
| RL Algorithm | S_Het | NS_Het | S_Pop | NS_Pop |
| ZIP $k = 1$ | 67.907 (1.150) | 73.830 (0.403) | 74.898 (1.016) | 78.651 (0.556) |
| ZIP $k = 4$ | 68.865 (1.067) | 73.836 (0.464) | 75.933 (1.114) | 80.413 (0.629) |
| ZIP $k = N$ | 69.448 (1.201) | 74.580 (0.475) | 76.312 (1.122) | 80.424 (0.648) |
| BLR $k = 1$ | 65.600 (1.139) | 70.703 (0.457) | 70.915 (1.024) | 74.782 (0.596) |
| BLR $k = 4$ | 68.045 (1.122) | 73.322 (0.505) | 75.766 (1.097) | 79.809 (0.622) |
| BLR $k = N$ | **69.757 (1.171)** | **75.393 (0.427)** | **77.272 (1.096)** | **81.675 (0.583)** |

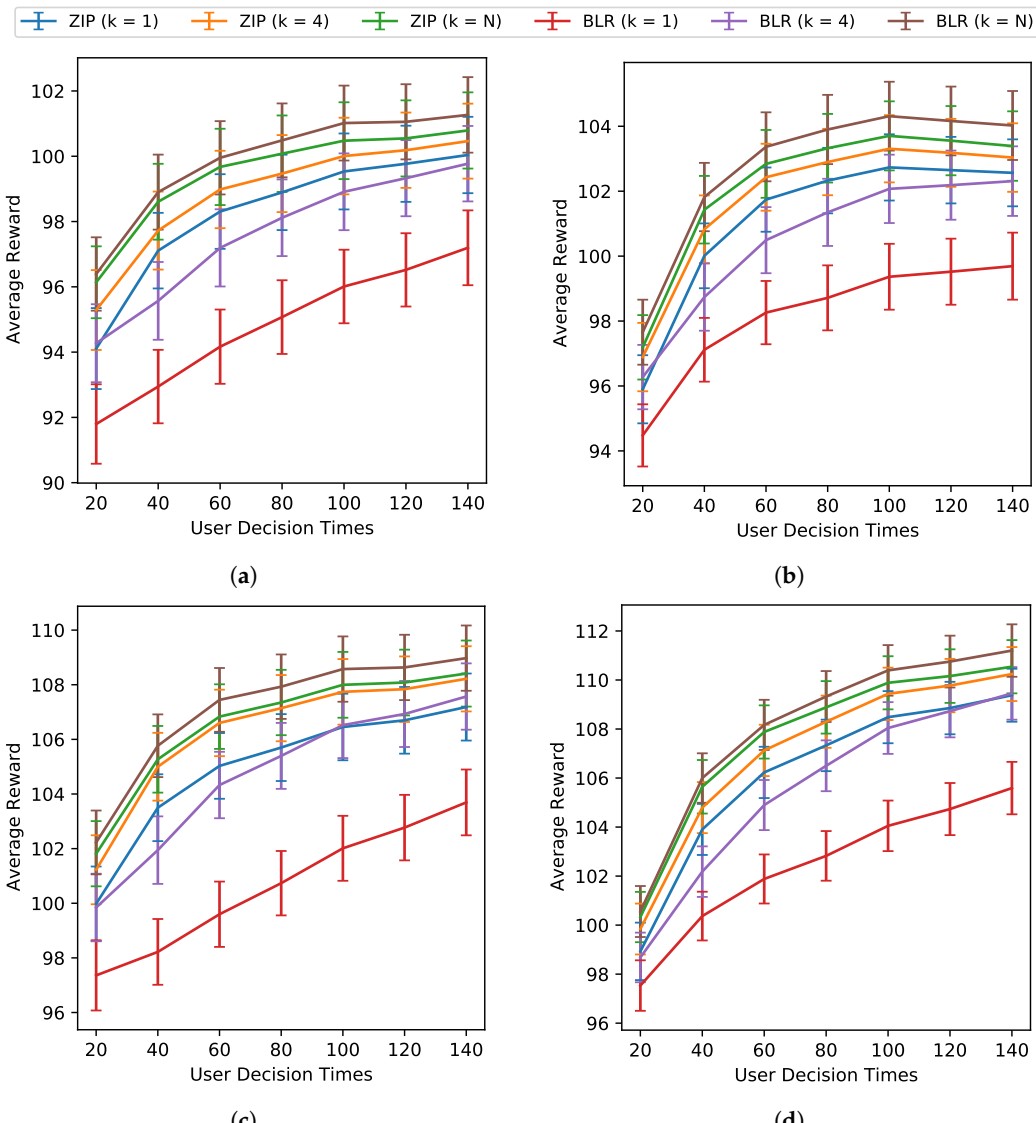

**Figure 3.** Average User Rewards Over Time. Above, we show simulation results of the six candidate algorithms (BLR and ZIP respectively for different cluster sizes $k$) across the four simulation environments. The y-axis is the mean and $\pm 1.96 \cdot$ standard error of the average user rewards $(\bar{R} = \frac{1}{72} \sum_{i=1}^{72} \frac{1}{t_0} \sum_{s=1}^{t_0} R_{i,s})$ for decision times $t_0 \in [20, 40, 60, 80, 100, 120, 140]$ across 100 Monte Carlo simulated trials. Standard error is $\frac{\hat{\sigma}}{\sqrt{100}}$ where $\hat{\sigma}$ is the sample variance of the 100 $\bar{R}$s. (**a**) Stationary Base Model and Heterogeneous Effect Size; (**b**) Nonstationary Base Model and Heterogeneous Effect Size; (**c**) Stationary Base Model and Population Effect Size; (**d**) Nonstationary Base Model and Population Effect Size.

2. **Cluster Size:** RL algorithms with larger cluster sizes $k$ perform better overall, especially for simulation environments with population-level treatment effects (rather than heterogeneous treatment effects). At first glance, one might think that algorithms with smaller cluster sizes may perform better because they can learn more personalized policies for users (especially in environments with heterogeneous treatment effects). Interestingly, though, the algorithms with larger cluster sizes performed better across all environments in terms of average reward (average across users and time) and the 25th-percentile of the average reward (average over time) across users (Table 2); this means that the RL algorithm candidates with larger cluster sizes performed better for both the average user and for the worst off users. The better performance of algorithms with larger cluster sizes is likely due to their ability to reduce noise and learn

faster by leveraging the data of multiple users to learn. Even though the algorithms with larger cluster sizes are less able to model and learn the heterogeneity across users, this is outweighed by the immense benefit of sharing data across users to learn faster and reduce noise.

There are some limitations to these experiments.

Fixed Reward Noise Variance for BLR: The BLR algorithm includes a noise variance hyperparameter ($\eta^2$ in Equation (A3)). In our experiments, we set $\eta^2$ to the reward variance observed in the ROBAS 2 data set. Assuming that $\eta^2$ is known is unrealistic; in the future, we plan to learn $\eta^2$ along with other BLR algorithm parameters in the real study. The known value of $\eta^2$ could be a reason that BLR performed comparably to ZIP.

More Distinct and Complex Simulation Environments: We may not be looking widely enough across environment variants to find settings where these algorithms perform differently. With sufficient data per user in a highly heterogeneous user environment, we expect cluster size $k = 1$ to do the best. In future work, we aim to add simulation environments with greater heterogeneity and less noise to see if large cluster sizes still perform well, and we aim to create more complex simulation environment variants that are more distinct (e.g., environments where users may differ by heterogeneous demographic features like age and gender). Additionally, we want to impute state features of interest in the real study that were not present in the data set, such as phone engagement.

Additional RL Algorithm Candidate Considerations. We also aim to consider other axes for algorithm candidates such as algorithms with other update cadences (e.g., every night or biweekly) and algorithms with an informative prior. In initial simulations using algorithms with informative priors, we found that since the same (limited amount) of ROBAS 2 data was used to build both the simulation environment and the prior, the algorithms did not need to learn much to perform well. An open question is how to develop both simulation environments and informative priors in a realistic way using a limited amount of data. Finally, we will also explore additional design decisions such as how to carefully design the feature space for the RL algorithm.

These investigations will determine the final algorithm that goes into the actual study.

## 7. Discussion and Future Work

In this paper, we present the first extension of the PCS framework for designing RL algorithms in digital intervention settings. The case study demonstrates how to use the PCS framework to make design decisions and highlights our ongoing work in designing the Oralytics RL algorithm. This work helps fellow researchers understand and balance the benefits and drawbacks of certain aspects of the RL algorithm they use for their digital intervention studies.

**Author Contributions:** Conceptualization, A.L.T. and K.W.Z.; methodology, A.L.T., K.W.Z. and S.A.M.; software, A.L.T.; validation, A.L.T., K.W.Z., I.N.-S., V.S., F.D.-V. and S.A.M.; formal analysis, A.L.T. and K.W.Z.; investigation, A.L.T.; resources A.L.T., K.W.Z., I.N.-S., V.S., F.D.-V. and S.A.M.; data curation, A.L.T. and K.W.Z.; writing—original draft preparation, A.L.T. and K.W.Z.; writing—review and editing, A.L.T., K.W.Z., I.N.-S., V.S., F.D.-V. and S.A.M.; visualization, A.L.T.; supervision, F.D.-V. and S.A.M.; project administration, I.N.-S., V.S. and S.A.M.; funding acquisition, I.N.-S., V.S., F.D.-V. and S.A.M. All authors have read and agreed to the submitted version of the manuscript.

**Funding:** This research was funded by NIH grants IUG3DE028723, P50DA054039, P41EB028242, U01CA229437, UH3DE028723, and R01MH123804. KWZ is also supported by the National Science Foundation grant number NSF CBET–2112085 and by the National Science Foundation Graduate Research Fellowship Program under Grant No. DGE1745303. Any opinions, findings, and conclusions or recommendations expressed in this material are those of the author(s) and do not necessarily reflect the views of the National Science Foundation.

**Institutional Review Board Statement:** Not applicable.

**Informed Consent Statement:** Not applicable. Data is de-identified.

**Data Availability Statement:** The ROBAS 2 training data is available online at https://github.com/ROBAS-UCLA/ROBAS.2/blob/master/inst/extdata/robas_2_data.csv (accessed on 31 May 2022). The source code and all other supplementary resources are available online at https://github.com/StatisticalReinforcementLearningLab/pcs-for-rl (accessed on 1 June 2022).

**Acknowledgments:** We are grateful for the guidance and support of Wei Wei Pan, Jiayu Yao, Jessamyn Jackson, and Doug Ezra Morrison throughout this project.

**Conflicts of Interest:** The authors declare no conflict of interest. The funders had no role in the design of the study; in the collection, analyses, or interpretation of data; in the writing of the manuscript, or in the decision to publish the results.

## Abbreviations

The following abbreviations are used in this manuscript:

RL      reinforcement learning
BLR      Bayesian linear regression
ZIP      zero-inflated Poisson
MDP      Markov decision process
RMSE      root mean squared error

## Appendix A. Simulation Environments

*Appendix A.1. Baseline Feature Space of the Environment Base Models*

The ROBAS 2 dataset has a variety of features that we anticipate to be associated with brushing duration. These include the time of day (morning vs. evening), weekday vs. weekend, and summaries of the user's past brushing behavior. Together with domain experts in behavioral health and dentistry, we chose the following features to use to fit a model of the reward. Recall that the ROBAS 2 dataset only includes data under no intervention, so for now we are only fitting a model for the baseline reward model (i.e., the brushing duration under action $A_{i,t} = 0$). In Appendix A.5 we discuss how to model brushing duration under action 1.

1. Bias/Intercept Term $\in \mathbb{R}$
2. Time of Day (Morning/Evening) $\in \{0, 1\}$
3. Prior Day Total Brushing Duration (Normalized) $\in \mathbb{R}$
4. Weekend Indicator (Weekday/Weekend) $\in \{0, 1\}$
5. Proportion of Nonzero Brushing Sessions Over Past 7 Days $\in [0, 1]$
6. Day in Study (Normalized) $\in [-1, 1]$

We use these features to generate two types of base reward environments (Stationary and Non-Stationary). The Stationary model of the base environment uses the state function $g(S_{i,t}) \in \mathbb{R}^5$ that only includes the first five features above. The Non-Stationary model of the base environment uses state $g(S_{i,t}) \in \mathbb{R}^6$ that corresponds to all of the above features.

Normalization of State Features

We normalize features to ensure that state features are all in a similar range. The Prior Day Total Brushing Duration feature is normalized using z-score normalization (subtract mean and divide by standard deviation) and the Day in Study feature (originally in the range $[1, 28]$ since the study length of ROBAS 2 is 28) is normalized to be between $[-1, 1]$. Note that when generating rewards, Day in Study was normalized based on Oralytics' anticipated 10-week study length (range is still $[-1, 1]$).

$$\text{Normalized Total Brushing Duration in Seconds} = (\text{Brushing Duration} - 172)/118$$

$$\text{Normalized Day in Study When Fitting Model} = (\text{Day} - 14.5)/13.5$$

$$\text{Normalized Day in Study When Generating Rewards} = (\text{Day} - 35.5)/34.5$$

*Appendix A.2. Environment Base Model*

We consider three model classes: (1) a zero-inflated Poisson, (2) a hurdle model with a square root transform, and (3) a hurdle model with a log transform, and choose one out of these three model classes for each user. We define these three model classes below. Additionally, below $g(S)$ is the baseline feature vector of the current state defined in Appendix A.1, $w_{i,b}, w_{i,p}, w_{i,\mu}$ are user-specific weight vectors, $\sigma^2_{i,u}$ is the user-specific variance for the normal component, and $\text{sigmoid}(x) = \frac{1}{1+e^{-x}}$ is the sigmoid function.

(1) Zero-Inflated Poisson Model for Brushing Duration

$$Z \sim \text{Bernoulli}\left(1 - \text{sigmoid}(g(S)^T w_{i,b})\right)$$

$$Y \sim \text{Poisson}\left(\exp\left(g(S)^T w_{i,p}\right)\right)$$

$$\text{Brushing Duration in Seconds}: D = ZY$$

(2) Hurdle Model with Square Root Transform for Brushing Duration

$$Z \sim \text{Bernoulli}\left(1 - \text{sigmoid}\left(g(S)^T w_{i,b}\right)\right)$$

$$Y \sim \mathcal{N}\left(g(S)^T w_{i,\mu}, \sigma^2_{i,u}\right)$$

$$\text{Brushing Duration in Seconds}: D = ZY^2$$

Note that the nonzero component of this model, $Y^2$, can also be represented as a constant times a noncentral chi-squared, where the noncentrality parameter is the square of the mean of the normal distribution.

(3) Hurdle Model with Log Transform for Brushing Duration

$$Z \sim \text{Bernoulli}\left(1 - \sigma\left(g(S)^T w_{i,b}\right)\right)$$

$$Y \sim \text{Lognormal}\left(g(S)^T w_{i,\mu}, \sigma^2_{i,u}\right)$$

$$\text{Brushing Duration in Seconds}: D = ZY$$

Since we want to simulate brushing duration in seconds, we also round outputs of the hurdle models to the nearest whole integer. Notice that the zero-inflated Poisson model is a mixture model with a latent state. The Bernoulli draw $Z$ is latent and represents the user's intention to brush, and the Poisson models the user's brushing duration when they intend to brush (this is because the brush time can still be zero when the user intends to brush). On the other hand, the hurdle model provides a model for brushing duration conditional on whether the user brushed or not. The Bernoulli draw $Z$ in the hurdle model is observed.

Note that the hurdle model is used for the simulation environment only and *not* the RL algorithm. The hurdle model conditions on a collider (e.g., whether the person brushes their teeth), thus potentially leading to causal bias [44,45]. For example, consider an unobserved cause $U$, intervention $A$, whether the user brushed or not $Z$, brushing duration $D$, and a

directed acyclic graph with $A \rightarrow Z, U \rightarrow Z, U \rightarrow D$, and $Z \rightarrow D$. Then conditioning on collider $Z$ of treatment opens a pathway from $A$ to $D$ through $U$ [46]. Suppose in reality $A$ only impacts whether the user brushes their teeth but not the duration. Then, if we condition on $Z$ to evaluate the impact of $A$ on $D$, we may erroneously learn that $A$ impacts the duration of brushing. This makes the hurdle unsuitable as a model for an RL algorithm that aims to learn causal effects.

*Appendix A.3. Fitting the Environment Base Models*

We use ROBAS 2 data to fit the brushing duration model under action 0 (no message). For all model classes, we fit one model per user. All models were fit using MAP with a prior $w_{i,b}, w_{i,p}, w_{i,\mu} \sim \mathcal{N}(0, I)$ as a form of regularization because we have sparse data for each user. Weights were chosen by running random restarts and selecting the weights with the highest log posterior density.

Fitting Hurdle Models: For fitting hurdle models for user $i$, we fit the Bernoulli component and the nonzero brushing duration component separately. We use $D_{i,t}$ to denote the $i$th ROBAS 2 user's brushing duration in seconds at the $t$ time point. Set $Z_{i,t} = 1$ if the original observation $D_{i,t} > 0$ and 0 otherwise. We fit a model for this Bernoulli component. We then fit a model for the normal component to either the square root transform $Y_{i,t} = \sqrt{D_{i,t}}$ or to inverse-log transform $Y_{i,t} = \exp(D_{i,t})$ of the $i$th user's nonzero brushing duration.

Fitting Zero-Inflated Poisson Models: For the zero-inflated Poisson model, we jointly fit parameters for both the Bernoulli and the Poisson components. Since the brushing durations in the ROBAS 2 data were integer values, we did not have to transform the observation to fit the zero-inflated Poisson model.

The fitted parameters for the environment base models can be accessed at: https:// github.com/StatisticalReinforcementLearningLab/pcs-for-rl/tree/main/sim_env_data (accessed on 1 June 2022).

Selecting the Model Class for Each User

To select the model class for user $i$, we fit all three model classes using user $i$'s data from ROBAS 2. We then chose the model class that had the RMSE. Namely, we choose the model class with the lowest $L_i$, where:

$$L_i := \sqrt{\sum_{t=1}^{T} (D_{i,t} - \hat{\mathbb{E}}[D_{i,t}|S_{i,t}])^2}$$

Recall that $D_{i,t}$ is the brush time in seconds for user $i$ at decision time $t$. Definitions of $\hat{\mathbb{E}}[D_{i,t}|S_{i,t}]$ for each model class are specified below in Table A1.

**Table A1.** Definitions of $\hat{\mathbb{E}}[D_{i,t}|S_{i,t}]$ for each model class. $\hat{\mathbb{E}}[D_{i,t}|S_{i,t}]$ is the mean of user model $i$ fitted using data $\{(S_{i,t}, D_{i,t})\}_{t=1}^{T}$.

| Model Class | $\hat{\mathbb{E}}[D_{i,t} \mid S_{i,t}]$ |
|---|---|
| Zero-Inflated Poisson | $[1 - \text{sigmoid}(g(S_{i,t})^T w_{i,b})] \cdot \exp\left(g(S_{i,t})^T w_{i,p}\right)$ |
| Hurdle (Square Root) | $[1 - \text{sigmoid}(g(S_{i,t})^T w_{i,b})] \cdot \left[\sigma_{i,\mu}^2 + (g(S_{i,t})^T w_{i,\mu})^2\right]$ |
| Hurdle (Log) | $[1 - \text{sigmoid}(g(S_{i,t})^T w_{i,b})] \cdot \exp\left(g(S_{i,t})^T w_{i,\mu} + \frac{\sigma_{i,\mu}^2}{2}\right)$ |

**Table A2.** Definitions of $\widehat{\mathbb{E}}[D_{i,t}|S_{i,t}, D_{i,t} > 0]$ and $\widehat{\text{Var}}[D_{i,t}|S_{i,t}, D_{i,t} > 0]$ for each model class. $\widehat{\mathbb{E}}[D_{i,t}|S_{i,t}, D_{i,t} > 0]$ and $\widehat{\text{Var}}[D_{i,t}|S_{i,t}, D_{i,t} > 0]$ is the mean and variance of the nonzero component of user model $i$ fitted using data $\{(S_{i,t}, D_{i,t})\}_{t=1}^{T}$.

| Model Class | $\widehat{\mathbb{E}}[\mathbf{D_{i,t}}\|\mathbf{S_{i,t}}, \mathbf{D_{i,t}} > \mathbf{0}]$ |
|---|---|
| Hurdle (Square Root) | $\sigma_{i,u}^2 + (g(S_{i,t})^T w_{i,\mu})^2$ |
| Hurdle (Log) | $\exp(g(S_{i,t})^T w_{i,\mu} + \frac{\sigma_{i,u}^2}{2})$ |
| Zero-Inflated Poisson | $\frac{\exp(g(S_{i,t})^T w_{i,p})\exp(\exp(g(S_{i,t})^T w_{i,p}))}{\exp(\exp(g(S_{i,t})^T w_{i,p}))-1}$ |

| | $\widehat{\text{Var}}[D_{i,t}\|S_{i,t}, D_{i,t} > 0]$ |
|---|---|
| Hurdle (Square Root) | $g(S_{i,t})^T w_{i,\mu}^4 + 3\sigma_{i,u}^4 + 6\sigma_{i,u}^2(g(S_{i,t})^T w_{i,\mu})^2 - \widehat{\mathbb{E}}[R_{i,t}\|S_{i,t}, R_{i,t} > 0]^2$ |
| Hurdle (Log) | $(\exp(\sigma_{i,u}^2) - 1) \cdot \exp(2g(S_{i,t})^T w_{i,\mu} + \sigma_{i,u}^2)$ |
| Zero-Inflated Poisson | $\widehat{\mathbb{E}}[D_{i,t}\|S_{i,t}, D_{i,t} > 0] \cdot (1 + \exp(g(S_{i,t})^T w_{i,p}) - \widehat{\mathbb{E}}[D_{i,t}\|S_{i,t}, D_{i,t} > 0])$ |

Table A3 lists the number of model classes for all users in the ROBAS 2 study that we obtained after the procedure was run.

**Table A3.** Number of selected model classes for the Stationary and Non-Stationary environments.

| Model Class | Stationary | Non-Stationary |
|---|---|---|
| Hurdle with Square Root Transform | 9 | 7 |
| Hurdle with Log Transform | 9 | 8 |
| Zero-Inflated Poisson | 14 | 17 |

*Appendix A.4. Checking the Quality of the Simulation Environment Base Model*

Appendix A.4.1. Checking Moments

Using the chosen user-specific models, we simulate 100 trials. In each trial, for each user in ROBAS 2, we use their respective model to generate a data trajectory $(S_{i,t}, R_{i,t})_{t=1}^{56}$ (note that the ROBAS 2 study had two brushing windows per day for 28 days for a total of 56 brushing windows). We then compute the following metrics for each of the trials and averaged across trials:

1. Proportion of Missed Brushing Windows:

$$\frac{1}{N}\sum_{i=1}^{N}\frac{1}{T}\sum_{t=1}^{T}\mathbb{I}[D_{i,t} = 0]$$

2. Average Nonzero Brushing Duration:

$$\frac{1}{N}\sum_{i=1}^{N}\frac{1}{\sum_{t=1}^{T}\mathbb{I}[D_{i,t} > 0]}\sum_{t=1}^{T}\mathbb{I}[D_{i,t} > 0]D_{i,t}$$

3. Variance of Nonzero Brushing Durations:
   Let $\widehat{\text{Var}}(\{X_k\}_{k=1}^{K})$ represent the empirical variance of $X_1, X_2, ..., X_K$.

$$\widehat{\text{Var}}(\{D_{i,t} : t \in [1\!:\!T], D_{i,t} > 0\}_{i=1}^{N})$$

4. Variance of Average User Brushing Durations:
   This metric measures the degree of between-user variance in average brushing.

$$\widehat{\text{Var}}\left(\left\{\frac{1}{T}\sum_{t=1}^{T}D_{i,t}\right\}_{i=1}^{N}\right)$$

5. Average of Variances of Within-User Brushing Durations:
   This metric measures the average amount of within-user variance.

$$\frac{1}{N}\sum_{i=1}^{N}\widehat{\mathrm{Var}}\big(\{D_{i,t}\}_{t=1}^{T}\big)$$

The base models slightly overestimate the proportion of missed brushing windows in the ROBAS 2 data set. Our base models also slightly underestimate the average brushing duration. Our base models also for the most part slightly overestimate the between-user and within-user variance of rewards.

**Table A4.** Comparing Moments Between Base Models and ROBAS 2 Data Set. Above, we use BDs to abbreviate Brushing Durations. Values for the Stationary and Nonstationary base models are averaged across 100 trials.

| Metrics | ROBAS 2 | Stationary | Non-Stationary |
|---|---|---|---|
| Proportion of Missed Brushing Windows | 0.376674 | 0.403114 | 0.397812 |
| Average Nonzero BDs | 137.768129 | 131.308445 | 134.676955 |
| Variance of Nonzero BDs | 2326.518304 | 2392.955018 | 2253.177853 |
| Variance of Average User BDs | 1415.920148 | 1699.126897 | 1399.615330 |
| Average of Variances of Within-User BDs | 1160.723506 | 1405.944459 | 1473.239769 |

Appendix A.4.2. Measuring If a Base Model Captures the Variance in the Data

We measure how well the fitted base models captured (1) whether or not the user brushed and (2) the variance of the brush time when the users did brush. To measure point (1) for each user model $i$, we calculate the statistic:

$$U_i := \frac{1}{T}\sum_{t=1}^{T}\left(\frac{\mathbb{I}[D_{i,t}>0]-\widehat{\mathbb{E}}[\mathbb{I}[D_{i,t}>0]|S_{i,t}]}{\widehat{\mathrm{Var}}[\mathbb{I}[D_{i,t}>0]|S_{i,t}]}\right)^2 \tag{A1}$$

where $\widehat{\mathbb{E}}[\mathbb{I}[D_{i,t}>0]|S_{i,t}] = 1 - \mathrm{sigmoid}(S_{i,t}^T w_{i,b})$ and $\widehat{\mathrm{Var}}[\mathbb{I}[D_{i,t}>0]|S_{i,t}] = \widehat{\mathbb{E}}[\mathbb{I}[D_{i,t}>0]|S_{i,t}] \cdot \mathrm{sigmoid}(S_{i,t}^T w_{i,b})$.

To measure point (2) for each user model $i$, we calculate the statistic:

$$U_i := \frac{1}{\sum_{t=1}^{T}\mathbb{I}[D_{i,t}>0]}\sum_{t=1}^{T}\mathbb{I}[D_{i,t}>0]\left(\frac{D_{i,t}-\widehat{\mathbb{E}}[D_{i,t}|S_{i,t},D_{i,t}>0]}{\widehat{\mathrm{Var}}[D_{i,t}|S_{i,t},D_{i,t}>0]}\right)^2 \tag{A2}$$

Definitions of $\widehat{\mathbb{E}}[D_{i,t}|S_{i,t},D_{i,t}>0]$ and $\widehat{\mathrm{Var}}[D_{i,t}|S_{i,t},D_{i,t}>0]$ for the nonzero component of each model class are specified in Table A2. For a user model to capture the variance in the data, $U_i$ should be close to 1. We calculate the empirical mean $\overline{U} = \frac{1}{N}\sum_{i=1}^{N}U_i$ and standard deviation $\overline{\sigma_U} = \mathrm{std}(U_i)$, and the approximate 95% confidence interval is $\overline{U} \pm 1.96 \times \frac{\overline{\sigma_U}}{\sqrt{N}}$.

Results are in Table A5. We can see that after computing the statistic for each user, the confidence interval is close to 1. We understand that the confidence intervals do not contain 1, which implies that we are overestimating the amount of variance in $\mathbb{I}[D>0]$ and underestimating the amount of variance in $D|D>0$. In the future, we hope to improve upon this statistic by considering nonlinear components in our base models.

**Table A5.** Statistic $U_i$ for Capturing Variance in the Data. Values are rounded to the nearest 3 decimal places.

| Metric | Stationary | Non-Stationary |
|---|---|---|
| Equation (A1) $\overline{U}$ | 0.811 | 0.792 |
| Equation (A1) $\overline{\sigma_U}$ | 0.146 | 0.150 |
| Equation (A1) Confidence Interval | (0.760, 0.861) | (0.739, 0.844) |
| Equation (A2) $\overline{U}$ | 3.579 | 3.493 |
| Equation (A2) $\overline{\sigma_U}$ | 4.861 | 4.876 |
| Equation (A2) Confidence Interval | (1.895, 5.263) | (1.803, 5.182) |

*Appendix A.5. Imputing Treatment Effect Sizes for Simulation Environments*

Recall that the ROBAS 2 dataset does not have any data under action 1 (send a message). Thus, to model the reward under action 1, we must impute.

Appendix A.5.1. Treatment Effect Feature Space

The following choice of the treatment effect (advantage) feature space was made after discussion with domain experts of which features are most likely to interact with the intervention (action). The Stationary model uses state $h(S) \in \mathbb{R}^4$ corresponding to the following features:

1. Bias/Intercept Term $\in \mathbb{R}$
2. Time of Day (Morning/Evening) $\in \{0, 1\}$
3. Prior Day Total Brushing Duration (Normalized) $\in \mathbb{R}$
4. Weekend Indicator (Weekday/Weekend) $\in \{0, 1\}$

The Non-Stationary model uses state $h(S) \in \mathbb{R}^5$ corresponding to all of the above features as well as the following:

5. Day in Study (Normalized) $\in \mathbb{R}$

Appendix A.5.2. Imputation Approach

For the zero-inflated Poisson model, we impute treatment effects on both the user's intent to brush (Bernoulli component) and the user's brushing duration when they intend to brush (Poisson component). Similarly, for the hurdle models, we impute treatment effects on both whether the user's brushing duration is zero (Bernoulli component) and the user's brushing duration when the duration is nonzero.

After incorporating effect sizes, brushing duration under action $A$ in state $S$ is $D$ where:

$$Z \sim \text{Bernoulli}(1 - \text{sigmoid}(g(S)^T w_{i,b} + A \times h(S)^T \Delta_{i,B}))$$

$$D = \begin{cases} ZY^2, Y \sim \mathcal{N}(g(x)^T w_{i,\mu} + A \times h(x)^T \Delta_{i,N}, \sigma_u^2) & \text{for hurdle square root} \\ Z\exp(Y), Y \sim \mathcal{N}(g(x)^T w_{i,\mu} + A \times h(x)^T \Delta_{i,N}, \sigma_{i,u}^2) & \text{for hurdle log normal} \\ ZY, Y \sim \text{Poisson}(\exp(g(x)^T w_{i,p} + A \times h(x)^T \Delta_{i,N})) & \text{for zero-inflated Poisson} \end{cases}$$

$\Delta_{i,B}, \Delta_{i,N}$ are user-specific effect sizes; we will also consider population-level effect sizes (same across all users), which we denote as $\Delta_B, \Delta_N$. $g(S)$ is the baseline feature vector as described in Appendix A.1, and $h(S)$ is the feature vector that interacts with the effect size as specified above.

Appendix A.5.3. Heterogeneous versus Population-Level Effect Size

We consider realistic heterogeneous effect sizes (each user has a unique effect size) and a realistic population-level effect size (all users who share the same base model class also share the same effect size).

Population-Level Effect Sizes: Recall that for the stationary base model we fit models for $Y$ and $Z$, and get user-specific parameters $w_{i,b}, w_{i,p} \in \mathbb{R}^5$ (zero-inflated Poisson model) and parameters $w_{i,b}, w_{i,\mu} \in \mathbb{R}^5$ (hurdle models). Values of the fitted parameters can be accessed at: https://github.com/StatisticalReinforcementLearningLab/pcs-for-rl/tree/main/sim_env_data (accessed on 1 June 2022). We use these parameters to form the population effect sizes as follows:

Zero-Inflated Models' Effect Sizes:

- $\Delta_B = \mu_{B,\text{avg}}$ where $\mu_{B,\text{avg}} = \frac{1}{4} \sum_{d \in [2:\,5]} \frac{1}{N} \sum_{i=1}^{N} |w_{i,b}^{(d)}|$.

- $\Delta_N = \mu_{N,\text{avg}}$ where $\mu_{N,\text{avg}} = \frac{1}{4} \sum_{d \in [2:\,5]} \frac{1}{N} \sum_{i=1}^{N} |w_{i,p}^{(d)}|$.

Hurdle Models' Effect Sizes:

- $\Delta_B = \mu_{B,\text{avg}}$ where $\mu_{B,\text{avg}} = \frac{1}{4} \sum_{d \in [2:\,5]} \frac{1}{N} \sum_{i=1}^{N} |w_{i,b}^{(d)}|$.

- $\Delta_N = \mu_{N,\text{avg}}$ where $\mu_{N,\text{avg}} = \frac{1}{4} \sum_{d \in [2:\,5]} \frac{1}{N} \sum_{i=1}^{N} |w_{i,\mu}^{(d)}|$.

We use $w_{i,b}^{(d)}, w_{i,p}^{(d)}, w_{i,\mu}^{(d)}$ to denote the $d^{\text{th}}$ dimension of the vector $w_{i,b}, w_{i,p}, w_{i,\mu}$ respectively; we take the minimum over all dimensions excluding $d = 1$, which represents the weight for the bias/intercept term.

Heterogeneous Effect Sizes: To calculate the heterogeneous effect sizes, we again group users by their chosen base model (zero-inflated, hurdle square-root, hurdle log). We then draw effect sizes for each user from a normal distribution specific to their base model:

$$\Delta_{i,B} \sim \mathcal{N}(\Delta_B, \sigma_B^2)$$

$$\Delta_{i,N} \sim \mathcal{N}(\Delta_N, \sigma_N^2)$$

$\Delta_B, \Delta_N$ are set to the population-level effect sizes for that base model class as described above. To set $\sigma_B^2, \sigma_N^2$, we do the following:

Zero-Inflated Models:

- $\sigma_B$ is the empirical standard deviation over $\{\mu_{i,B}\}_{i=1}^{N}$ where $\mu_{i,B} = \frac{1}{4} \sum_{d \in [2:\,5]} |w_{i,b}^{(d)}|$.

- $\sigma_N$ is the empirical standard deviation over $\{\mu_{i,N}\}_{i=1}^{N}$ where $\mu_{i,N} = \frac{1}{4} \sum_{d \in [2:\,5]} |w_{i,p}^{(d)}|$.

Hurdle Models:

- $\sigma_B$ is the empirical standard deviation over $\{\mu_{i,B}\}_{i=1}^{N}$ where $\mu_{i,B} = \frac{1}{4} \sum_{d \in [2:\,5]} |w_{i,b}^{(d)}|$.

- $\sigma_N$ is the empirical standard deviation over $\{\mu_{i,N}\}_{i=1}^{N}$ where $\mu_{i,N} = \frac{1}{4} \sum_{d \in [2:\,5]} |w_{i,\mu}^{(d)}|$.

After the procedure described above, we set $\sigma_B = 0.192$ for the hurdle models, $\sigma_B = 0.193$ for the zero-inflated model, $\sigma_N = 0.576$ for the hurdle square-root model, $\sigma_N = 0.173$ for the hurdle log model, and $\sigma_N = 0.163$ for the zero-inflated model (values are rounded to the nearest 3 decimal places). Histograms of $\Delta_{i,B}, \Delta_{i,N}$ and values of $\mu_B, \mu_N$ for each base model class are specified in Figure A1.

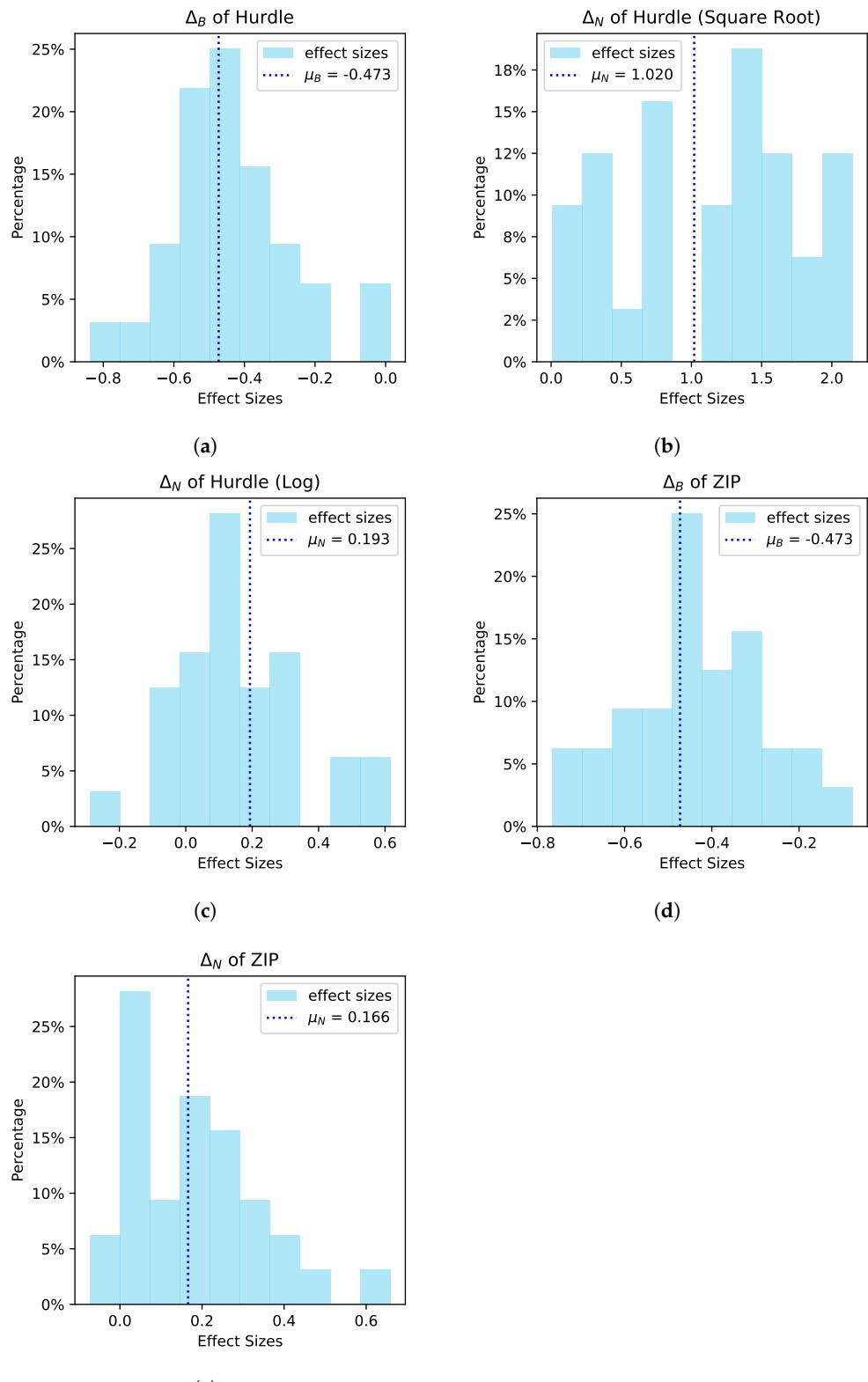

**Figure A1.** Effect sizes $\Delta_{i,B}$'s, $\Delta_{i,N}$'s, $\mu_B$, $\mu_N$ for each base model class. The effect sizes are used to generate rewards under action $A = 1$ for the simulation environment. (**a**) Bernoulli component (hurdle); (**b**) Nonzero component (square root); (**c**) Nonzero component (log); (**d**) Bernoulli component (ZIP); (**e**) Poisson component (ZIP).

## Appendix B. RL Algorithm Candidates

*Appendix B.1. Feature Space for the RL Algorithm Candidates*

We use $f(S) \in \mathbb{R}^3$ to denote the feature space used by our RL algorithm candidates to predict the advantage (i.e., the immediate treatment effect). $f(S)$ contains the following features:

1. Bias/Intercept Term $\in \mathbb{R}$
2. Time of Day (Morning/Evening) $\in \{0, 1\}$
3. Prior Day Total Brushing Duration (Normalized) $\in \mathbb{R}$

The normalization procedure for Prior Day Brushing Duration is the same as the one described in Appendix A.1.

We use $m(S) \in \mathbb{R}^4$ to denote the feature space used by the RL algorithm candidates to approximate the baseline reward function. $m(S)$ contains all the above features as well as the following:

4. Weekend Indicator (Weekday/Weekend) $\in \{0, 1\}$

Note that the feature space used by the RL algorithm candidates is different than the feature space used to model the reward in the simulation environments, specified in Appendices A.1 and A.5; this means that the RL algorithms will have a misspecified reward model. Namely, the baseline feature space for the simulation environment has an additional Proportion of Nonzero Brushing Sessions Over Past 7 Days feature and the non-stationary variant has the Day in Study (Normalized) feature. The treatment effect feature space for the simulation environment has an additional Weekend Indicator (Weekday/Weekend) and the non-stationary variant has the Day in Study (Normalized) feature.

The rationale for not including the Day in Study (Normalized) feature is although we wanted to capture potential non-stationarity in brushing outcomes in order to create a realistic simulation environment, our RL algorithm candidates do not have reward functions that vary arbitrarily over time. We do not include Proportion of Nonzero Brushing Sessions Over Past 7 Days and Weekend Indicator (Weekday/Weekend) to detect the robustness of RL algorithm candidates to a misspecified reward model.

*Appendix B.2. Decision 1: Reward Approximating Function*

The first decision in designing the RL algorithm is the choice between using a linear model or a zero-inflated Poisson model as the reward approximating function used by the posterior sampler (note that this is separate from the reward model used to generate the environment). More information on how the posterior sampling algorithm performs action selection can be found in Appendix C.2. Appendix C.1 provides information about how the algorithm updates at update times.

Note that the function $m$ for the RL algorithm's baseline reward model is only used at update times. The function $f$ for the RL algorithm's advantage model is used at both decision and update times.

Appendix B.2.1. Bayesian Linear Regression Model

The first candidate is to use the following reward generating model with action centering used in [1] for the posterior sampler:

$$R_{i,t} = m(S_{i,t})^T \alpha_{i,0} + \pi_{i,t} f(S_{i,t})^T \alpha_{i,1} + (A_{i,t} - \pi_{i,t}) f(S_{i,t})^T \beta_i + \epsilon_{i,t} \tag{A3}$$

where $\alpha_{i,0} \in \mathbb{R}^4$ and $\alpha_{i,1}, \beta_i \in \mathbb{R}^3$. $\pi_{i,t}$ is the probability that action $A_{i,t} = 1$ is selected by the RL algorithm for user $i$ in state $S_{i,t}$; we discuss how to compute this probability in Appendix C.2. The RL algorithm models $\epsilon_{i,t}$ as being drawn from $\mathcal{N}(0, \eta^2)$ (the choice of $\eta^2$ is informed by the ROBAS 2 dataset). Additionally, we put uninformative normal priors on the parameters: $\alpha_{i,0} \sim \mathcal{N}(0, \sigma_{prior} I_4)$, $\alpha_{i,1} \sim \mathcal{N}(0, \sigma_{prior} I_3)$, $\beta_i \sim \mathcal{N}(0, \sigma_{prior} I_3)$, where $\sigma_{prior} = 5$.

Appendix B.2.2. Zero-Inflated Poisson Regression Model

The second candidate is to use the zero-inflated reward generating model for the posterior sampler:

$$
\begin{aligned}
Z_{i,t} &\sim \text{Bernoulli}\left(1 - \text{sigmoid}(m(S_{i,t})^T \alpha_{i,b} + A_{i,t} \cdot f(S_{i,t})^T \beta_{i,b})\right) \\
Y_{i,t} &\sim \text{Poisson}\left(\exp\left(m(S_{i,t})^T \alpha_{i,p} + A_{i,t} \cdot f(S_{i,t})^T \beta_{i,p}\right)\right) \\
R_{i,t} &= Z_{i,t} Y_{i,t}
\end{aligned}
\tag{A4}
$$

The above model closely resembles the zero-inflated Poisson model class used to develop the simulation environment in Appendix A.2; however, recall that the feature space used by the RL algorithm and the model to generate the environment is different. Additionally, here, we directly model the reward, rather than the raw brushing duration.

Additionally, the posterior sampling algorithm will put the following uninformative normal priors on the parameters: $\alpha_{i,b}, \alpha_{i,p} \sim \mathcal{N}(0, \sigma_{prior} I_4)$ and $\beta_{i,b}, \beta_{i,p} \sim \mathcal{N}(0, \sigma_{prior} I_3)$, where $\sigma_{prior} = 5$.

*Appendix B.3. Decision 2: Cluster Size*

Clustering involves grouping $k$ users together and pooling all their data together for the RL algorithm. This gives us one RL algorithm instantiation per cluster (no data shared across clusters). For our experiments, we draw $N = 72$ simulated users (the expected sample size for the Oralytics study) with replacement and cluster these users at random (every possible cluster is equally likely). We then keep these cluster assignments fixed across the trials.

For simplicity in running our experiments, we consider randomly formed clusters, but we are thinking of clustering by entry date in the real study. Recall that we want to cluster users who are similar to each other. A natural approach is to cluster users by a baseline feature, but we cannot predict how many users who share the same baseline feature will join within a relatively short period of time (e.g., we cannot depend on there being four females within the first two weeks). Entry date is a reasonable clustering criterion because domain experts believe that users who enter the study around the same time will be similar. Users who enter near the end of the study may be very different from users who enter near the beginning because of societal factors (e.g., pandemic restrictions being lifted), seasonal influences (e.g., differences in a user's mood in spring and midwinter), and fidelity (e.g., quality of onboarding procedures and staff experience may improve over time). One natural approach is to cluster by baseline features, but that is not feasible for a study with a slow recruitment rate, like Oralytics.

**Appendix C. RL Algorithm Posterior Updates and Posterior Sampling Action Selection**

*Appendix C.1. Posterior Updates to the RL Algorithm at Update Time*

During the update step, the reward approximating function will update the posterior with newly collected data. Additionally, we make $M$ draws of the parameters from the updated posterior and use them for all decision times until the next update time. Here are the procedures for how the Bayesian linear regression model and the zero-inflated Poisson model perform posterior updating.

Appendix C.1.1. Bayesian Linear Regression Model

Suppose we are selecting actions for decision time $t$. Let $\phi(S_{i,t}, A_{i,t}) = [m(S_{i,t}), \pi_{i,t} f(S_{i,t}), (A_{i,t} - \pi_{i,t}) f(S_{i,t})]$ be the joint feature vector and $\theta_i = [\alpha_{i,0}, \alpha_{i,1}, \beta_i]$ be the joint weight vector. Notice that Equation (A3) can be vectorized in the form: $R_{i,t} = \phi(S_{i,t}, A_{i,t})^T \theta_i + \epsilon_{i,t}$. Now let $\Phi_{i,1:t-1}$ be the matrix of all stacked vectors $\{\phi(S_{i,s}, A_{i,s})\}_{s=1}^{t-1}$, and $\mathbf{R}_{i,1:t-1}$ be a vector of stacked rewards $\{R_{i,s}\}_{s=1}^{t-1}$, where we have batch data of the $t - 1$ decision times before the current update time.

Recall that we have normal priors on $\theta_i$ where $\theta_i \sim \mathcal{N}(\mu_{\text{prior}}, \Sigma_{\text{prior}})$, where $\mu_{\text{prior}} = \mathbf{0} \in \mathbb{R}^{3+3+4}$ and $\Sigma_{\text{prior}} = \text{diag}(\sigma^2_{prior}I_3, \sigma^2_{prior}I_3, \sigma^2_{prior}I_4)$. The posterior distribution of the weights given current history $H_{i,t-1}$, $p(\theta_i|H_{i,t-1})$ is conjugate and is also normal.

$$\theta_i|H_{i,t-1} \sim \mathcal{N}(\mu^{posterior}_{i,t-1}, \Sigma^{posterior}_{i,t-1})$$

$$\Sigma^{posterior}_{i,t-1} = \left( \frac{1}{\eta^2} \Phi^T_{i,1:t-1} \Phi_{i,1:t-1} + \Sigma^{-1}_{prior} \right)^{-1}$$

$$\mu^{posterior}_{i,t-1} = \Sigma^{posterior}_{i,t-1} \left( \frac{1}{\eta^2} \Phi^T_{i,1:t-1} \mathbf{R}_{i,1:t-1} + \Sigma^{-1}_{prior} \mu_{prior} \right)$$

Note that we fit $\eta^2$ to the ROBAS 2 dataset and fixed it for all of our experiments. For the real study, we are considering assigning a conjugate prior on $\eta^2$ and updating it at update times.

Appendix C.1.2. Zero-Inflated Poisson Regression Model

For the zero-inflated Poisson regression model, the posterior distribution of the weights $\theta_i = \{\alpha_{i,b}, \beta_{i,b}, \alpha_{i,p}, \beta_{i,p}\}$ given data $H_{i,t-1}$, $p(\theta_i|H_{i,t-1})$ does not have a closed form. Therefore, we use Metropolis-Hastings (MH) with a normal proposal distribution as an approximate posterior sampling method.

Posterior Density:

The log-likelihood of the zero-inflated Poisson regression model is:

$$\log f(R_{i,t}|S_{i,t}, A_{i,t}; \theta_i) = \begin{cases} \log((1-p) + p \exp(-\lambda)) & R = 0 \\ \log p - \lambda + R \log \lambda - \log R! & R = 1, 2, 3, ... \end{cases}$$

where $p = 1 - \text{sigmoid}(m(S_{i,t})^T \alpha_{i,b} + A_{i,t} \cdot f(S_{i,t})^T \beta_{i,b})$ is the probability of the user intending to brush, and $\lambda = \exp(m(S_{i,t})^T \alpha_{i,p} + A_{i,t} \cdot f(S_{i,t})^T \beta_{i,p})$ is the expected Poisson count.

Therefore, the log posterior density is:

$$\log p(\theta_i|H_{i,t-1}) \propto \sum_{n=1}^{N} \log f(R_{i,t}|S_{i,t}, A_{i,t}; \theta_i) + \log p(\theta_i)$$

Proposal Distribution:

We choose a normal distribution for our proposal distribution. At each step of MH, we propose a new sample given the old sample, $\theta^k_{\text{prop}} \sim \mathcal{N}(\theta^k_{\text{old}}, \gamma^2 I)$, where $\theta^k$ denotes the $k$th value of $\theta$.

Metropolis-Hastings Acceptance Ratio:

The Metropolis-Hastings acceptance ratio given a proposed sample $\theta_{\text{prop}}$ and an old sample $\theta_{\text{old}}$ is defined as:

$$\alpha(\theta_{\text{prop}}, \theta_{\text{old}}) := \min\left(1, \frac{p(\theta_{\text{prop}})/q(\theta_{\text{prop}}|\theta_{\text{old}})}{p(\theta_{\text{old}})/q(\theta_{\text{old}}|\theta_{\text{prop}})}\right)$$

Since our proposal distribution is symmetric, the log acceptance ratio becomes:

$$\log \alpha(\theta_{\text{prop}}, \theta_{\text{old}}) := \min(0, \log p(\theta_{\text{prop}}) - \log p(\theta_{\text{old}}))$$

*Appendix C.2. Action Selection at Decision Time*

Our action selection scheme at decision time selects action $A_{i,t} \sim \text{Bern}(\pi_{i,t})$ where $\pi_{i,t} = \text{clip}(\tilde{\pi}_{i,t})$. clip is the clipping function defined in Appendix C.2.2 and $\tilde{\pi}_{i,t}$ is the posterior probability that $A_{i,t} = 1$ is optimally defined in Appendix C.2.1.

Appendix C.2.1. Posterior Sampling

Bayesian Linear Regression Model: Based on the Bayesian linear regression model of the reward, specified by Equation (A3):

$$\tilde{\pi}_{i,t} = \text{Pr}_{\tilde{\beta} \sim \mathcal{N}(\mu_{i,t-1}^{post}, \Sigma_{i,t-1}^{post})} \left\{ f(S_{i,t})^T \tilde{\beta} > 0 \big| S_{i,t}, H_{i,t-1} \right\}$$

Note that the randomness in the probability above is only over the draw of $\tilde{\beta}$ from the posterior distribution.

Zero-Inflated Poisson Model: Based on the zero-inflated Poisson model of the reward, specified by Equation (A4):

$$\tilde{\pi}_{i,t} = \text{Pr}_{\tilde{\alpha}_{i,b}, \tilde{\alpha}_{i,p}, \tilde{\beta}_{i,b}, \tilde{\beta}_{i,p}} \left\{ \tilde{Z}_{i,t} \tilde{Y}_{i,t} > 0 \big| S_{i,t}, H_{i,t-1} \right\}$$

where $\tilde{Z}_{i,t} \sim \text{Bernoulli}\big(1 - \text{sigmoid}(m(S_{i,t})^T \tilde{\alpha}_{i,b} + A_{i,t} \cdot f(S_{i,t})^T \tilde{\beta}_{i,b})\big)$ and $\tilde{Y}_{i,t} \sim \text{Poisson}\big(\exp\big(m(S_{i,t})^T \tilde{\alpha}_{i,p} + A_{i,t} \cdot f(S_{i,t})^T \tilde{\beta}_{i,p}\big)\big)$. Note that the randomness in the probability above is only over the draw of $(\tilde{\alpha}_{i,b}, \tilde{\alpha}_{i,p}, \tilde{\beta}_{i,b}, \tilde{\beta}_{i,p})$ from the posterior distribution.

Appendix C.2.2. Clipping to Form Action Selection Probabilities

Since we want to facilitate after-study analyses, we clip action selection probabilities using the action clipping function for some $\pi_{\min}, \pi_{\max}$ where $0 < \pi_{\min} \leq \pi_{\max} < 1$ is chosen by the scientific team:

$$\text{clip}(\pi) = \min(\pi_{\max}, \max(\pi, \pi_{\min})) \in [\pi_{\min}, \pi_{\max}] \tag{A5}$$

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
