# Peer review of "Designing Reinforcement Learning Algorithms for Digital Interventions: Pre-Implementation Guidelines"

_algorithms, doi:10.3390/a15080255_

Round 1

Reviewer 1 Report

Well written manuscript. Overall contents are very focused and pleased to read. A little grammatical glitches, but they are very minor.

Author Response

Thank you so much for taking the time to read our paper and for commenting that the manuscript is well-written and a pleasure to read. Thank you for pointing out the grammatical errors. We have fixed the grammatical errors and revised the sentence structure. Please see the attachment below containing the marked-up version of the manuscript.

Reviewer 2 Report

This manuscript performed well, showing real-time Reinforcement Learning in Digital Interventions. Since every smart system implements very strict control loads and rules due to the current remote control revolution, so this could service is actually a very important issue for remote smart system control.

This research work is a good reminder to the design of RL algorithms for the digital interventions setting and connection cloud computation that these are the biggest contribution of this research. Some suggestions and comments please careful to correct the content of the manuscript.

1.     The system figure 1 is a decision times and actions for the RL algorithm in the Oralytics study; please add some explanations in these parts. Please describe the execution steps and procedures of the system in more detail to confirm whether there are any problems.

2.     In figure 3, please describe the execution steps and procedures of the system in more detail to confirm whether there are any problems?

3.     Please add the meaning of the diagrams in figure 2.

4.     Be careful to write the references again. Some error formats must be correct.

Author Response

Thank you so much for taking the time to read our manuscript and for your thoughtful comments.  We really appreciate that you thought that the manuscript performed well. Please see the attachment below containing the marked-up version of the manuscript. In addition, below are the changes we made in response to your feedback:

  1. In the caption of Figure 1 (pg. 3 in the marked-up version), we added in specific Oralytics examples of the decision times, the action space, and the reward. We provided more information on the study procedure's execution steps, describing the decisions that the RL algorithm makes for a user over the course of the study. We also modified the figure and added the specific state, action, and reward definitions for Oralytics.
  2. In the caption of Figure 3 (pg. 16 in the marked-up version), we detailed each step in the procedure we used to calculate the metric across time to generate the graphs in the figure. Note that we ran 50 more trials per simulation for a total of 100 trials to reduce the error bars.
  3. In the caption of Figure 2 (pg. 12 in the marked-up version), we provided more  background information about the ROBAS 2 study and summarized the histogram's key takeaways.
  4. Thank you for pointing out formatting errors in the references. We were not sure if you were referring to the related works section or the bibliography. We checked both. We checked that each citation is correctly formatted and used in the paper. In  the "Related Works" section (starting on pg. 7 in the marked-up version) we changed all references from first author surname to first author surname plus "et al". If we have not addressed your formatting concerns, please let us know and we will make the necessary changes.

Reviewer 3 Report

The authors of the paper presented an approach to designing reinforcement learning algorithms for online applications. They showed a framework for helping to make design decisions in those methods and applied it to a mobile health intervention system that was designed for oral self-care behaviors.

The article is very good and shows valuable insights. Honestly, I am pretty sure that it is not the very first version of this paper and maybe it was reviewed earlier.

I have only two comments.

Firstly, I think that numerical results should be described in a more extensive way because the description leaves room for speculation and insinuation.

Secondly, the font in the figures is way too small (minor).

Author Response

Thank you for taking the time to read our manuscript and providing thoughtful comments. We appreciate that you believe the article shows valuable insights and that you recognize the effort we put into working on the paper. Please see the attachment below containing the marked-up version of the manuscript. In regards to your feedback:

  1. We agree that the numerical results could indeed have been presented more clearly. We changed the paper to describe the experiment procedure and our analysis of the results more precisely. (pp. 14 - 18 in the marked-up version) We also made structural changes by moving the discussion of the experiment's takeaways earlier in the section and clearly separating our discussions of the takeaways and the limitations of the experiments.
  2. Thank you for pointing out that the font in the figures is too small. We have increased the font size in all figures for easier readability.

Round 2

Reviewer 2 Report

Questions asked by reviewers are answered